# Computational imaging during video game playing shows dynamic synchronization of cortical and subcortical networks of emotions

**Joana Leitão**[1,2]*, **Ben Meuleman**[2], **Dimitri Van De Ville**[3,4], **Patrik Vuilleumier**[1,2]

**1** Laboratory for Behavioral Neurology and Imaging of Cognition, Department of Fundamental Neuroscience, University of Geneva, Geneva, Switzerland, **2** Swiss Center for Affective Sciences, University of Geneva, Geneva, Switzerland, **3** Institute of Bioengineering, Center for Neuroprosthetics, École Polytechnique Fédérale de Lausanne (EPFL), Geneva, Switzerland, **4** Department of Radiology and Medical Informatics, University of Geneva, Geneva, Switzerland

* joana.leitao@unige.ch

**Data Availability Statement:** All relevant data are available in the Supporting information files and in

## Abstract

Emotions are multifaceted phenomena affecting mind, body, and behavior. Previous studies sought to link particular emotion categories (e.g., fear) or dimensions (e.g., valence) to specific brain substrates but generally found distributed and overlapping activation patterns across various emotions. In contrast, distributed patterns accord with multi-componential theories whereby emotions emerge from appraisal processes triggered by current events, combined with motivational, expressive, and physiological mechanisms orchestrating behavioral responses. According to this framework, components are recruited in parallel and dynamically synchronized during emotion episodes. Here, we use functional MRI (fMRI) to investigate brain-wide systems engaged by theoretically defined components and measure their synchronization during an interactive emotion-eliciting video game. We show that each emotion component recruits large-scale cortico-subcortical networks, and that moments of dynamic synchronization between components selectively engage basal ganglia, sensory-motor structures, and midline brain areas. These neural results support theoretical accounts grounding emotions onto embodied and action-oriented functions triggered by synchronized component processes.

## Introduction

Emotions are pervasive phenomena that promote adaptive responses to behaviorally relevant events. However, the functional and neuroanatomical organization of emotion is still unresolved. What are the essential neural circuits coordinating the complex, multiple, and often abrupt changes in both mental and bodily states that are characteristically associated with emotion? Do they rely on specialized modules or distributed systems in the brain, and which are these? Such questions have been hotly debated in past decades [1]. Affective neuroscience approaches have generally focused on theoretical models postulating the existence of distinct emotion categories (e.g., fear, joy) or dichotomous dimensions (e.g., valence, arousal)

the Neurovault database (https://identifiers.org/neurovault.collection:8740).

**Funding:** This study was funded by the National Center of Competence in Research (NCCR) Affective Sciences (51NF40-104897) and a Sinergia grant from the Swiss National Science Foundation (CRII5-180319). The funders had no role in study design, data collection and analysis, decision to publish, or preparation of the manuscript.

**Competing interests:** The authors have declared that no competing interests exist.

**Abbreviations:** ACC, anterior cingulate cortex; BOLD, blood oxygenation level dependent; CPM, Component Process Model; dmPFC, dorsomedial prefrontal cortex; EDA, electrodermal activity; ELSA, Emergent Liquid State Affect; EMG, electromyography; EPI, echoplanar image; fMRI, functional MRI; GLM, general linear model; GM, gray matter; HRF, hemodynamic response function; mPFC, medial prefrontal cortex; OFC, orbitofrontal cortex; PCC, posterior cingulate cortex; PFC, prefrontal cortex; RM-ANOVA, repeated-measures ANOVA; RNN, recurrent neural network; RR, respiration rate; STAI, State-Trait Anxiety Inventory; STS, superior temporal sulcus; SVD, singular value decomposition; vmPFC, ventromedial prefrontal cortex.

associated with dedicated brain circuits (e.g., fear is assumed to be processed in the amygdala, disgust in the insula) [2, 3]. Yet accumulating evidence suggests that overlapping and distributed activation patterns emerge across different emotion categories, regardless of valence or arousal differences (e.g., insula activates to pain and pleasure, amygdala to fear and humor) [4, 5]. These findings advocate for taking a broader system approach to go beyond a simple one-to-one mapping between emotions and brain regions and thus provide a more comprehensive account of their functional richness and intricacy.

Alongside discrete and dimensional accounts of emotion, classic psychology theories proposed componential models of emotions within an appraisal framework. According to these models, emotions are multi-componential processes involving time-varying changes in appraisal mechanisms that encode contextual information about an event (e.g., goal conduciveness, coping potential, etc.), action tendencies (approach, avoidance, etc.), expressive behaviors (gestures, vocalizations, etc.), as well as changes in autonomic bodily function and subjective feelings [6–8]. An influential componential theory is the Component Process Model (CPM; Fig 1), in which the different emotion components are assumed to rely on distinct but coordinated subsystems [7, 9]. This model characterizes emotions as dynamic interactions between components, whose combination gives rise to the emergent subjective feeling quality of a particular emotional episode when a certain level of transient synchronization occurs between components. As such, componential models of emotions fit well with recent neurobiological models that envision brain functions in relation to large-scale functional networks with ongoing interactive activity, and might thus offer a fruitful framework to explain the co-occurring and widespread brain activation patterns typically observed across different emotions [5]. Nevertheless, despite solid foundation and extensive research in psychology [10–12], componential models remain poorly understood at the brain level and still neglected in neuroscience.

Investigating such models would require going beyond traditional emotion elicitation paradigms. Most neuroimaging studies employ paradigms based on a third-person view-point with static or indirect stimuli (e.g., faces, voices) without any active interaction or personal relevance for the viewer, which are likely to probe emotion perception or recognition more than actual generative processes underlying self-experienced emotions. Additionally, in line with assumptions from discrete or dimensional theories, neuroimaging studies typically compare predefined categories of emotions (e.g. fear versus disgust, or pleasant versus unpleasant), rather than more general appraisal components that may be jointly engaged across different emotions but to different degrees (e.g., goal-obstructive events coupled with low coping potential could elicit fear, while goal-conducive events in high coping potential contexts could elicit satisfaction) [9, 10].

Hence, the goals of our study were 3-fold. First, we aimed at creating a new task that allowed manipulating different appraisals within an interactive and self-relevant environment. Second, we used functional MRI (fMRI) to delineate distinct functional brain networks associated with the Appraisal, Motivation, Expression and Physiology components as postulated by the CPM [10] (Fig 1). Focusing on these 4 nonexperiential components offers a comprehensive characterization of emotional episodes that goes beyond the classical reliance on self-reported measures of feeling states along predefined categorical labels [12]. Third, based on the notion that emotional episodes may emerge from coordinated pattern organization across components, we derived dynamic synchronization measures between emotion component and between corresponding brain networks, using both a computational modelling approach and data-driven fMRI analyses, in order to identify brain regions critical for orchestrating this synchronization.

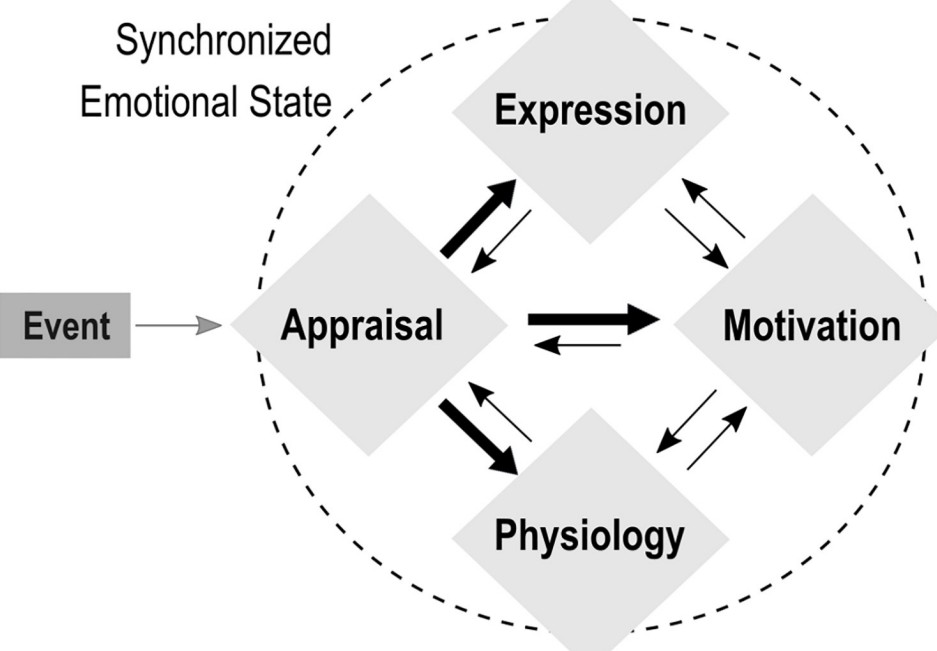

**Fig 1. Schematic depiction of the CPM.** Following a relevant event, components of Appraisal, Motivation, Physiology, and Expression dynamically interact through feedforward (thick arrows) and feedback (thin arrows) processes, producing globally synchronized states that correspond to an emotional episode. CPM, Component Process Model.

To do so, we designed an arcade game in which we manipulated both Goal Conduciveness and Coping Potential Appraisals while we recorded brain activity in participants with fMRI, as well as physiological signals with peripheral sensors and facial expressive behaviors with electromyography (EMG). We could thus obtain time-resolved measures for each emotion component as postulated in the CPM, including Appraisal processes through changes in Goal Conduciveness and Coping Potential across different game situations, Motivation through participants' behavior in these situations, Expression through facial EMG, and Physiology through peripheral measures. Our results unveil that emotion components each recruit distributed networks of cortical and subcortical brain areas that have been previously associated with cognitive and affective processes. Critically, a restricted set of core regions is distinctively engaged during moments of dynamic synchronization between the 4 components, including basal ganglia, sensory-motor structures, and medial areas in prefrontal and posterior cingulate cortex. These results support the long-debated idea that emotional responses may be grounded in embodied and action-based representations, coupled with higher-order functions associated with self-reflective processes, presumably underpinned by these core brain regions.

## Results

### Inducing emotions through appraisal of game variables

Our first goal was to create a task allowing the investigation of emotions in a direct, self-relevant context. As the CPM assumes that appraisals constitute an essential trigger of emotional episodes, our task explicitly manipulated the occurrence of particular self-relevant game situations instead of manipulating predefined emotions, as traditionally done in neuroimaging

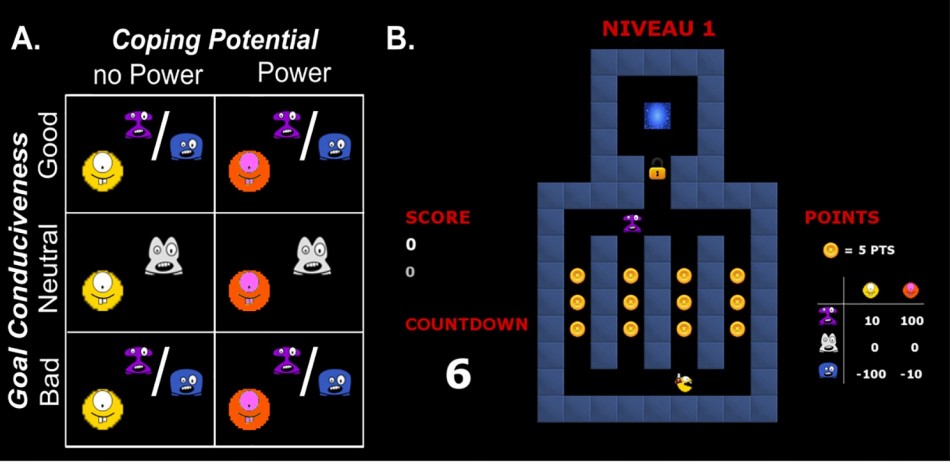

**Fig 2. Illustration of the interactive video game.** (A) Experimental conditions defined by a 2 × 3 design manipulating Appraisals of (i) Goal Conduciveness (bad, neutral, good monsters) and (ii) Coping Potential (no-power, power). The colors of bad and good monsters were counterbalanced across participants. (B) Illustration of the maze interface in one trial. The maze configuration and colors varied across trials within each condition.

studies. Hence, we designed an arcade game that manipulated 2 Appraisals whose role has been solidly established in psychology research: Goal Conduciveness and Coping Potential (Fig 2). Participants controlled a yellow avatar that navigated in a maze in which coins had to be collected. Across different levels corresponding to our Appraisal manipulations, they could encounter one of 3 types of "monsters" that produced different outcomes when touched: (i) good monsters yielded 10 points to the participant, (ii) neutral monsters yielded no points, and (iii) bad monsters caused a loss of 100 points. These different monster types thus manipulated the Appraisal of Goal Conduciveness, since the overall goal of the game was to collect as many points as possible.

Orthogonally, in half the levels, a magic potion cue was added to the avatar and gave the possibility to activate a "super-power." Once activated by the participant, the super-power changed the outcome of touching the monsters: good monsters now yielded 100 points, while bad monsters took only 10 points from the participant. The outcome of neutral monsters remained unaltered (providing a baseline condition). The super-power option therefore manipulated the Appraisal of Coping Potential.

Each trial corresponded to specific levels of Goal Conduciveness and Coping Potential combinations, during which participants navigated in the maze together with monsters and collected as many points as possible. After 8 s, a door would open, prompting participants to go to a teleporter zone that would take them to the next trial. Once the door opened, participants had a limited amount of time (countdown period) to reach the teleporter, or they would lose all the points gained in the current trial.

Participants played 3 runs of this game inside the fMRI scanner while we recorded peripheral autonomic variables and facial EMG to measure the Physiology and Expression components, respectively. The Motivation component was assessed through behavioral measures recorded during the game (see below), providing indices for approach and avoidance tendencies.

### Appraisal manipulations produce expected effects on subjective ratings

To assess the effectiveness of our Appraisal manipulations and their impact on subjective experience, we obtained ratings about appraisals and affective feelings during an additional run

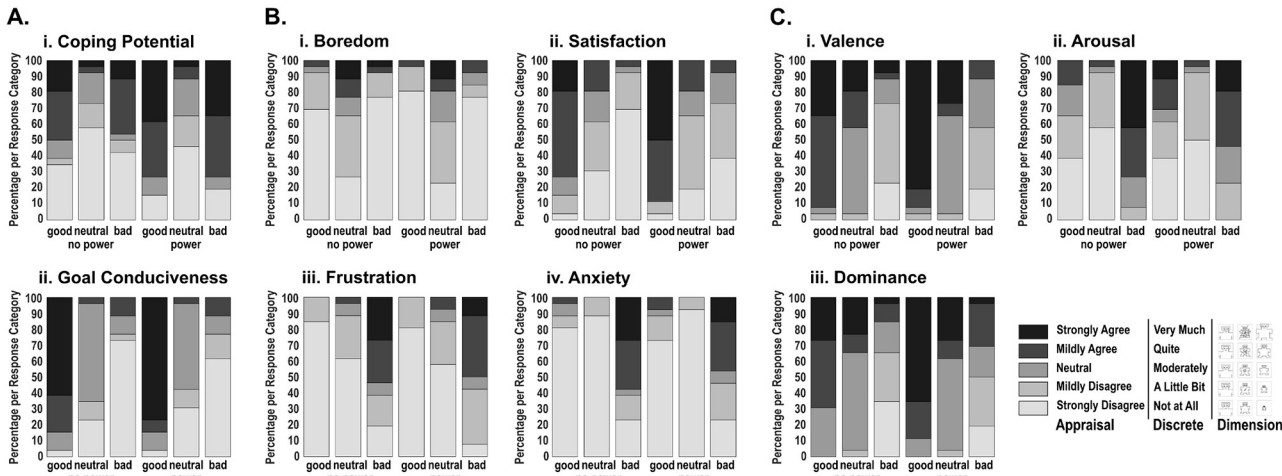

**Fig 3. Response frequencies across participants for subjective ratings (post scan), using 5-point Likert scales (see Methods).** Response labels used for the different questions are displayed on the right lower panel. (A) Ratings for Appraisals of (i) Coping Potential and (ii) Goal Conduciveness. (B) Discrete emotion ratings for (i) Boredom, (ii) Satisfaction, (iii) Frustration, and (iv) Anxiety. (C) Dimensional emotion ratings for (i) Valence, (ii) Arousal, and (iii) Dominance. Data used in this figure can be found in S1 Data.

outside the scanner, using screenshots of different conditions encountered in the game (see Methods).

Regarding Coping Potential, participants evaluated neutral monsters as the condition in which they had the least power over outcomes, regardless of their actual power status (Fig 3A.i). With good and bad monsters, judgments of being able to modify outcomes were more prevalent in the power compared to no-power conditions, indicating that this Appraisal was modulated as intended. Our manipulation of Goal Conduciveness was also effective: touching monsters was perceived as more congruent with participants' goals for good than bad monsters, while the neutral monster condition was intermediate (Fig 3A.ii).

Emotion feeling ratings were also queried using both discrete and dimensional labels. Among discrete emotions, participants reported more boredom with neutral, higher satisfaction with good, and higher anxiety or frustration with bad monsters (Fig 3B). Along dimensional aspects, participants rated the good and bad monster conditions as the most and least positively valenced, respectively (Fig 3C.i). They also felt calmer with neutral and more aroused with bad monsters, while dominance was rated weak during bad conditions but strong during good conditions (Fig 3C.ii and 3C.iii). Altogether, these data validate an effective manipulation of Appraisals with our game design and confirm a reliable elicitation of different emotional experiences in participants.

## Emotion components recruit distinctive brain-wide networks

Our second goal was to delineate brain networks recruited by the different emotion components as postulated by the CPM. The Appraisal network was identified using a standard general linear model (GLM) analysis of fMRI data based on the experimental conditions manipulated in our game, whereas networks modulated by Motivation, Physiology, and Expression components were determined using a data-driven GLM based on the behavioral, peripheral physiological, and EMG recordings obtained during the task, respectively.

**Appraisal networks.** To unveil brain networks engaged by Appraisal processes, we computed the main effects of Coping Potential and Goal Conduciveness as well as their

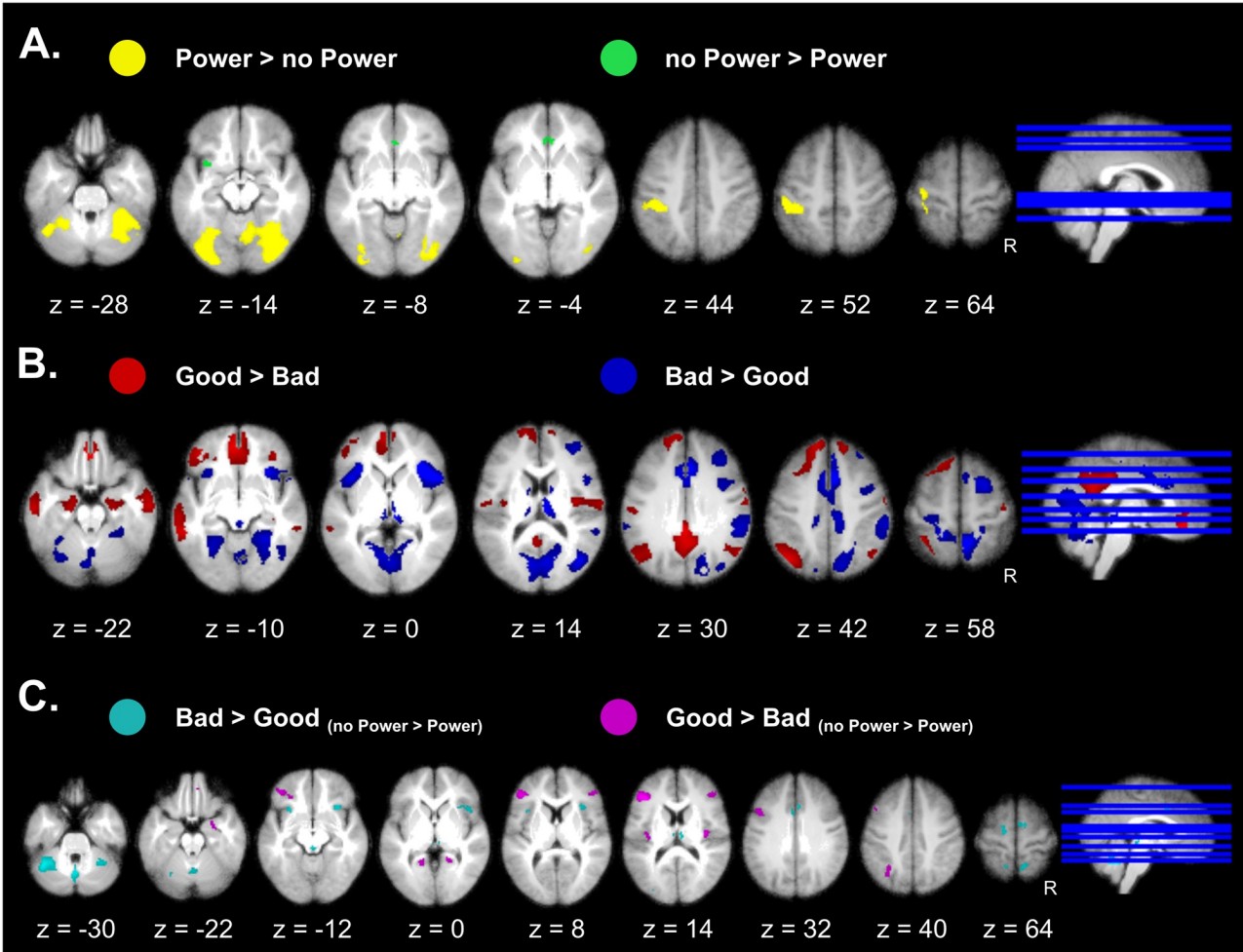

**Fig 4. Component brain networks engaged by Appraisal processes.** (A) Effects of Coping Potential showing effects of high and low power. (B) Effects of Goal Conduciveness showing the differential effects between good and bad monsters. (C) Interaction effects between Coping Potential and Goal Conduciveness. Effects are presented on axial slices of a mean brain image created by averaging the participants' normalized structural images, with a statistical threshold of $p_{FWE} < 0.05$. Individual beta maps used for this figure are available on Neurovault at https://identifiers.org/neurovault.collection: 8740.

interactions using a repeated-measures ANOVA (RM-ANOVA) based on a block design. For the main effect of Coping Potential, trials with power relative to no-power (pooled across levels of Goal Conduciveness) evoked higher activation in sensory-motor areas and cerebellum, as well as extrastriate visual cortex (Fig 4A, Table A in S1 Table). This accords with a role of these areas in motor action control and planning functions (Fig F in S1 Fig), presumably modulated by the different navigation strategies used by participants across power conditions. Please note that keypresses were regressed out from analyses and cannot account for these activations. The opposite comparison showed higher activation in the anterior cingulate cortex (ACC; area BA33) and insula for the no-power conditions (Fig 4A, Table A in S1 Table), consistent with a role of these regions in uncertainty monitoring [13–15] and lower strategic control in this condition.

Main effects of Goal Conduciveness were identified by comparing "good versus neutral" and "bad versus neutral" conditions (pooled across levels of Coping Potential). Both comparisons showed distributed activations (Fig A in S1 Fig) that overlapped in bilateral fronto-

parietal areas, lateral extrastriate visual areas, striatum, and cerebellum, brain regions generally implicated in visuo-motor coordination, spatial navigation, and attention. Importantly, there were also marked differences between the 2 conditions. The good monsters produced higher activation in the amygdala and hippocampal regions bilaterally, ventromedial prefrontal cortex (vmPFC), and posterior cingulate cortex (PCC), whereas bad monsters activated the bilateral anterior insula, as well as the ACC, thalamus, posterior parahippocampal gyrus, and medial occipital cortex (Fig A in S1 Fig). These differences were still apparent in direct comparisons between good and bad monsters (Fig 4B, Table B in S1 Table), and reminiscent of networks often reported for positive/rewarding and negative/aversive stimuli [5]. While this activation pattern is consistent with the way Goal Conduciveness was manipulated here (through gain and losses) and previously reported effects of emotional valence (Fig G in S1 Fig), it is important to note that the Goal Conduciveness Appraisal is not uniquely defined by valence but involves other cognitive processes mediated by distributed regions, as highlighted by the current contrasts. More generally, these neural data further validate the effectiveness of our game by demonstrating a modulation of brain systems found to be responsive to emotional valence in other studies.

Finally, we tested for interactions between the 2 Appraisal processes, by computing the contrasts "bad$_{(no-power > power)}$ > good$_{(no-power > power)}$" and "good$_{(no-power > power)}$ > bad$_{(no-power > power)}$." Such interaction effects were selectively observed in the anterior insula, ACC, periaqueductal gray, and right amygdala, all areas associated with emotion processing and reactive behavior to salient stimuli (Fig 4C, Table C in S1 Table). Additional activations were observed in the dorsal and inferior prefrontal cortex (PFC), posterior insula/rolandic opercula, posterior parietal cortex, and cerebellum (Fig 4C, Table C in S1 Table), a pattern consistent with sensory-motor integration and action selection network [16, 17].

Overall, these findings support predictions from the CPM according to which different combinations of appraisals may produce distinct emotional responses, e.g., with goal-obstructive situations eliciting more aversive feelings when people have no power to control the situation, and vice versa for positive feelings in goal-congruent situations [18], consistent with modulations found in emotional brain networks here.

**Motivation network.** To identify the brain network associated with Motivation processes, we derived 4 behavioral indices of approach and avoidance tendencies from each participant's gameplay, namely (i) the average number of times participants were caught from the back, representing situations of avoidance (ApproachTail), or (ii) instead caught from the front, representing approach (ApproachHead), plus (iii) the time spent during countdown periods before reaching the teleporter zone (CountdownTime), and (iv) the number of coins collected in a given trial (EatenCoins), the 2 latter indices providing indirect measures of approach/avoidance, as well as engagement/reward seeking during the task.

These Motivation indices were carefully validated at the behavioral level (Table 1). First, the neutral condition was used as a baseline control and showed no differences across Coping Potential levels for any of these indices (two-sided paired $t$ tests; ApproachTail: $t(25) = 1.76$,

**Table 1. Behavioral measures indexing the Motivation component, averaged across participants (±SD).** Data used in this table can be found in S2 Data.

| Motivation Indices | No-Power | | | Power | | |
|---|---|---|---|---|---|---|
| | **Good** | **Neutral** | **Bad** | **Good** | **Neutral** | **Bad** |
| **No. ApproachTail** | 0.40 ± 0.20 | 0.04 ± 0.03 | 1.55 ± 0.35 | 0.48 ± 0.34 | 0.03 ± 0.03 | 1.62 ± 0.32 |
| **No. ApproachHead** | 3.96 ± 0.97 | 0.60 ± 0.11 | 1 ± 0.34 | 4.33 ± 1.21 | 0.62 ± 0.19 | 1.25 ± 0.53 |
| **CountdownTime (s)** | 3.58 ± 0.81 | 2.39 ± 0.40 | 1.84 ± 0.39 | 4.14 ± 0.82 | 2.41 ± 0.34 | 2.06 ± 0.44 |
| **No. EatenCoins** | 11.97 ± 1.98 | 15.63 ± 0.78 | 11.82 ± 1.17 | 9.44 ± 2.70 | 15.60 ± 0.79 | 12.17 ± 1.3 |

$p = 0.09$, $d_{av} = 0.42$; ApproachHead: $t(25) = -0.43$, $p = 0.67$, $d_{av} = 0.11$; EatenCoins: $t(25) = 0.66$, $p = 0.54$, $d_{av} = 0.23$; CountdownTime: $t(25) = -0.63$, $p = 0.53$, $d_{av} = 0.06$).

Next, we examined the effect of Appraisal manipulations on Motivation indices by computing 2 (Coping Potential: power, no-power) × 2 (Goal Conduciveness: good versus neutral, bad versus neutral) RM-ANOVAs for each of the 4 indices. For the ApproachTail index, we found main effects of Coping Potential ($F(1,25) = 16.15$; $p < 0.001$; $\eta_p^2 = 0.392$) and Goal Conduciveness ($F(1,25) = 600.05$; $p < 0.001$; $\eta_p^2 = 0.960$) but no interaction ($F(1,25) = 0.02$; $p = 0.876$; $\eta_p^2 = 0.001$). The latter main effect indicates more frequent events "caught from the back" with bad compared to good monsters, both during power and no-power (Table 1). The ApproachHead index also revealed main effects of Coping Potential ($F(1,25) = 22.04$; $p < 0.001$; $\eta_p^2 = 0.468$) and Goal Conduciveness ($F(1,25) = 123.38$; $p < 0.001$; $\eta_p^2 = 0.832$), and no interaction ($F(1,25) = 0.71$; $p = 0.408$; $\eta_p^2 = 0.028$), but now reflecting more frequent events "caught from the front" with good compared to bad monsters (Table 1). These behavioral data neatly demonstrate active approach behaviors when encountering good monsters in the game but avoidance of bad monsters. In addition, generally more frequent "caught" events with power than no-power may reflect increased risk behavior in power situations.

For CountdownTime, the RM-ANOVA revealed main effects of Coping Potential ($F(1,25) = 26.738$; $p < 0.001$; $\eta_p^2 = 0.517$) and of Goal Conduciveness ($F(1,25) = 115.15$; $p < 0.001$; $\eta_p^2 = 0.822$), and an interaction between the two ($F(1,25) = 10.066$; $p = 0.004$; $\eta_p^2 = 0.287$). This interaction was due to a longer time spent in the maze during the countdown period in power relative to no-power conditions, further increased with good compared to bad monsters (Table 1), indicating that participants felt less pressed to reach the teleporter when they were not threatened by bad monsters.

Finally, for EatenCoins, we found an effect of Coping Potential ($F(1,25) = 47.708$; $p < 0.001$; $\eta_p^2 = 0.905$) and a significant interaction ($F(1,25) = 13.244$; $p = 0.015$; $\eta_p^2 = 0.726$), characterized by fewer coins eaten in the power than no-power condition with good monsters but slightly more coins eaten in the power condition with bad monsters (Table 1). This reflected a change in the relevance of coins as a point-gaining source depending on the current condition. In the presence of good monsters, the value of eating coins decreased with power (as touching the monster was more beneficial than catching coins), whereas it increased in the presence of bad monsters (as points lost by touching the monster and points gained from the coins became almost similar). The main effect of Goal Conduciveness ($F(1,25) = 0.112$; $p = 0.75$; $\eta_p^2 = 0.022$) was not significant.

Altogether, these results clearly establish that participants exhibited strategical adjustments in their behaviors in response to different game situations defined by Appraisal manipulations and modified their navigation in the maze accordingly. Approach tendencies were more frequent with good monsters and avoidance more frequent with bad monsters, with power generally increasing the former but decreasing the latter. These findings not only further validate our game design but also provide precise quantitative indices to characterize Motivation processes in our task.

Each of these indices was therefore transformed into parametric modulator regressors, computed across all experimental conditions, and then entered into regression analyses of fMRI data to define brain activity maps that reflected these behavioral tendencies. To determine networks engaged across different dimensions of the Motivation component, we computed a single contrast pooling (i.e., summing) the parameter estimates of the 4 indices together. This revealed several areas in which activity varied (either increased or decreased) with changes in motivation indices, including the orbitofrontal cortex (OFC), the ACC, and

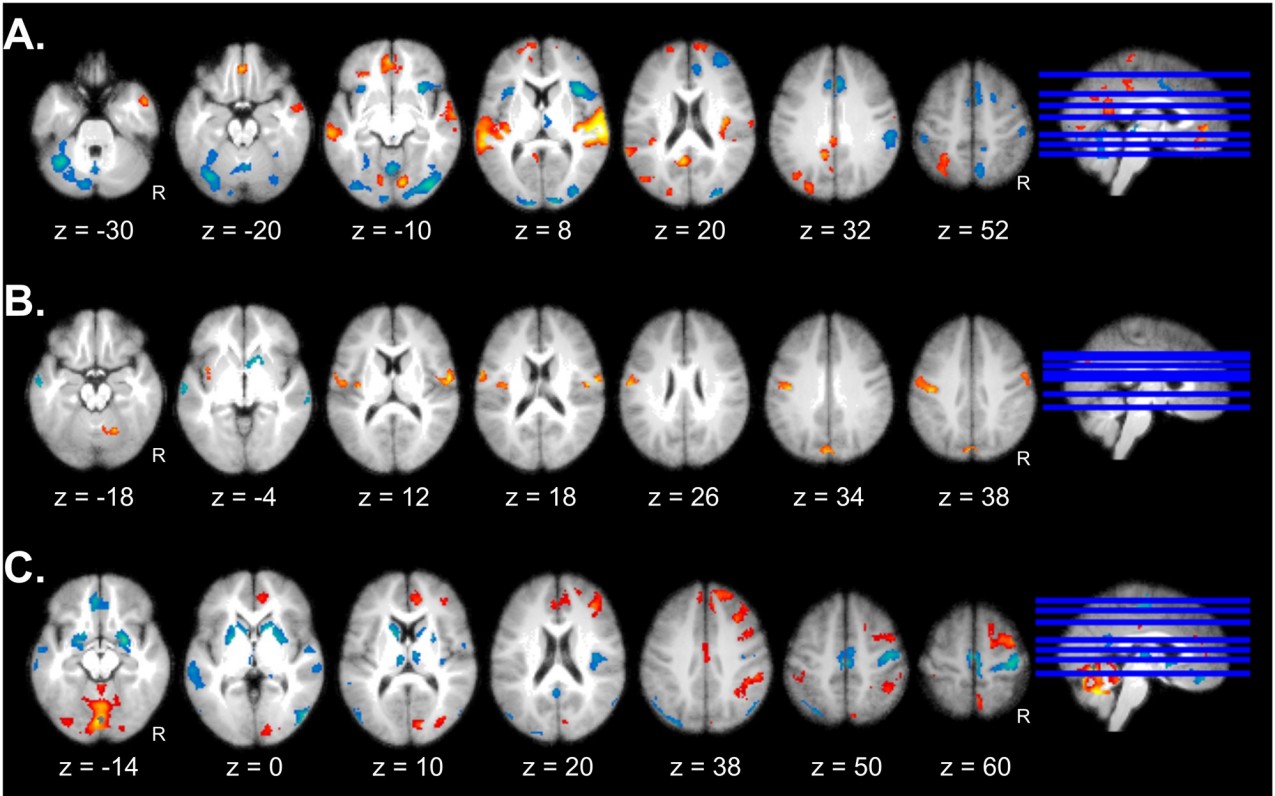

**Fig 5. Emotion component networks.** Component brain networks whose activity is modulated by Motivation (A), Expression (B), and Physiology (C). Red/orange colors reflect positive and blue colors reflect negative correlations with the respective indices. Effects are presented on axial slices of a mean image created by averaging the participants' normalized structural images and displayed at a voxel height threshold $p < 0.001$ with a cluster-level threshold of $p_{FWE} < 0.05$. Individual beta maps used for this figure are available on Neurovault at https://identifiers.org/neurovault.collection: 8740.

medial frontal areas (including the supplementary motor area), right dorsolateral PFC, bilateral insula, as well as bilateral inferior parietal lobes, cerebellum, and visual cortex (Fig 5A, Table D in S1 Table). These areas overlap with those frequently implicated in emotion processing and value-based decision making [5]. There were also effects in bilateral auditory areas, likely because several indices were associated with game events accompanied by sound effects (e.g., the countdown period comprised the corresponding spoken numbers).

**Expression network.** To identify brain networks modulated by the Expression component, we acquired EMG activity from the zygomaticus and corrugator facial muscles during gameplay and computed the averaged instantaneous amplitude time courses of muscular activity across the 2 muscles. These time courses were then convolved with the hemodynamic response function (HRF) and entered in fMRI regression analyses, similar to our procedure for Motivation (see earlier). Results showed that spontaneous changes in facial expressions modulated bilateral primary motor areas (overlapping with typical face somatotopy in motor cortex), as well as the cerebellum and cuneus (Fig 5B, Table E in S1 Table).

**Physiology network.** Finally, to delineate a network associated with the Physiology component, we acquired cardiac, respiratory, and electrodermal activity (EDA) measures throughout the game. The instantaneous respiration rate (RR; convolved with HRF) and phasic EDA (not convolved: see Methods) time courses were again correlated with fMRI data in regression analyses. Heart rate time courses were not included as they were too noisy to allow reliable analysis. Both RR and EDA were then combined into a common contrast in a second-level

ANOVA. Results revealed a distributed network of cortical and subcortical areas significantly modulated by these physiological indices, including amygdala, striatum, and thalamus, as well as the cingulate and orbitofrontal cortices, right PFC, posterior insula, bilateral temporal cortices, and cerebellum (Fig 5C, Table F in S1 Table).

## Emotion component synchronization

The above results highlight that each emotion component engages specific brain networks corresponding to neural systems previously associated with relevant affective functions and behaviors. Critically, according to the CPM [10], emotional responses are determined by dynamic and recurrent interactions between these components, such that emotional experiences and associated feelings evoked by particular situations reflect the emergent pattern of brain-wide activity generated by a transient synchronization between components. Based on this theoretical assumption, we employed 2 independent approaches to estimate moments of synchronization between emotion components and identify brain activity patterns associated with such synchronization.

**Synchronization identified by brain networks coordination.** In a first brain-based approach, we hinged on component networks delineated by our fMRI analyses above and derived measures of synchronization between them to pinpoint neural substrates selectively recruited during transient synchronization of the components. For each of the 4 componential networks, we calculated a representative time course of activity, for each participant and each run (Fig 6, Fig B in S1 Fig). These were determined by computing the time point by time point scalar product between brain maps obtained for a given component based on 2 training runs and the whole-brain blood oxygenation level dependent (BOLD) time courses of the third test run [19]. The resulting network time courses were then used to compute pairwise synchronization time courses based on instantaneous phase coherence, which yielded a rank 4 similarity matrix per time point (Fig 6, Fig B in S1 Fig). Multivariate synchronization time courses were obtained by calculating the spectral radius as the maximum eigenvalue of each of these similarity matrices, normalized by the sum of all eigenvalues. Finally, the time points corresponding to the 5% highest synchronization values across all participants were combined into a z-score map, representing brain areas whose activity was selectively increased during peaks of synchronization between emotion components (Fig 6, Fig B in S1 Fig).

Results from this brain-based synchronization analysis revealed activations in a sensorimotor and associative network comprising bilateral basal ganglia, posterior insula, somatosensory parietal areas, motor cortex, dorsomedial prefrontal cortex (dmPFC), and PCC, as well as visual occipital areas and temporal areas (Fig D in S1 Fig). Hence, this set of brain regions appeared to be distinctively recruited in moments of synchronization between the 4 networks mediating separate emotion components.

**Synchronization identified through computational modelling.** In parallel, to constrain our results with a "periphery" measure of synchronization, independent of our "central" synchronization index calculated from the brain data above, we employed a previously validated computational model [20] to define synchronization moments based on empirically defined (behavioral and peripheral) measures of each component during the gameplay. This computational model of emotion processing [20] uses liquid state machines to model nonlinear dynamic and recurrent interactions between the different component signals. Here, the synchronization index was computed as a latent variable over the model's liquid state transformations, representing continuous and recursive influences between peripheral measures of each component in the model [20] (see Methods for a more detailed description). For each participant, we convolved this synchronization time course with the HRF, added it as a further

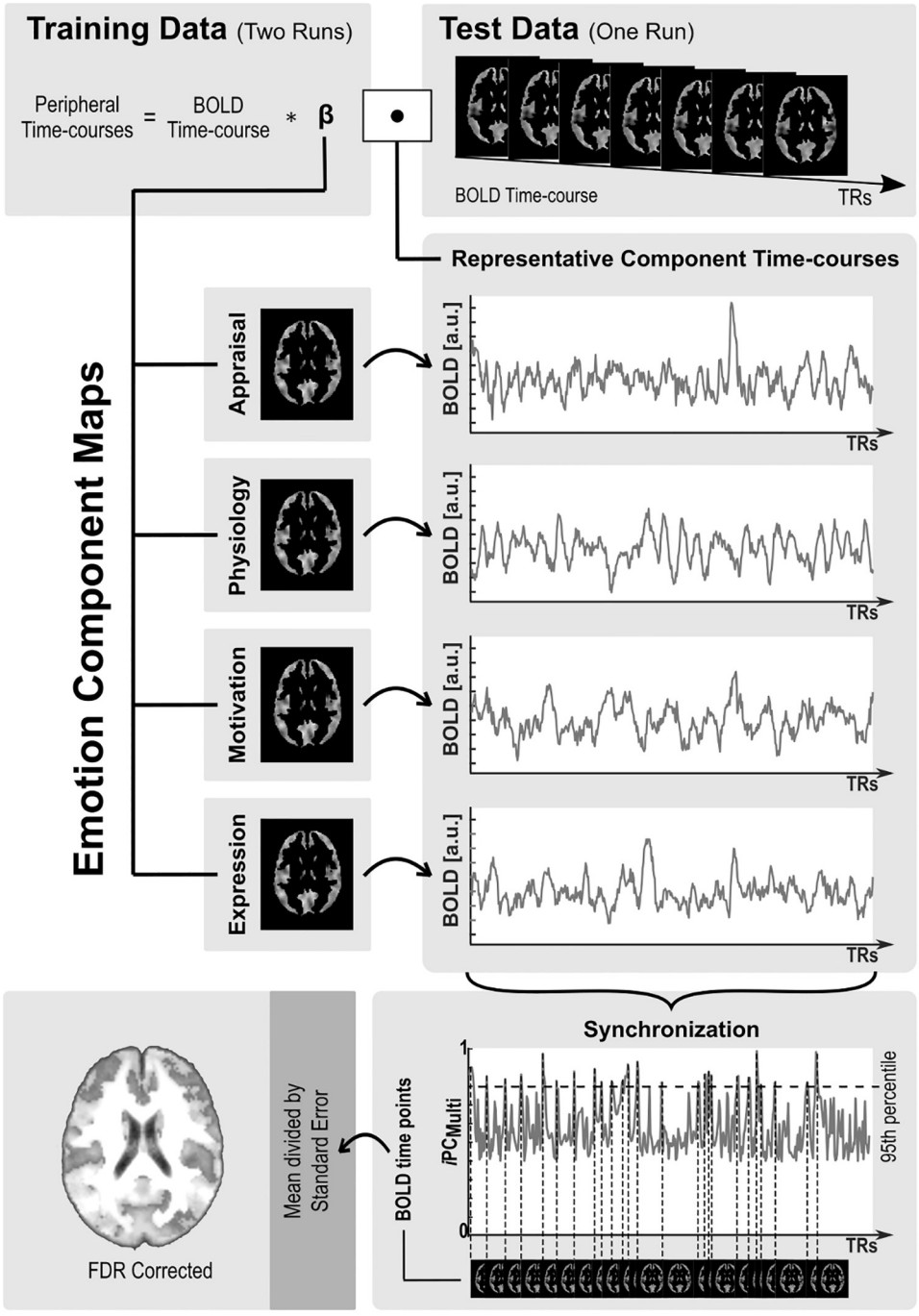

**Fig 6. Overview of the analysis pipeline for the brain-based calculation of synchronization between emotion component networks.** Representative time courses of emotion component networks were computed from the time point by time point scalar product between the emotion component maps learned for the training set and the BOLD time course of the test set. Synchronization between all 4 representative network time courses was estimated using a multivariate version of the instantaneous phase coherence as a similarity metric. Brain maps of regions selectively activated during high synchronization between all components were obtained by computing z-score across all BOLD volumes that were associated with 5% of the highest synchronization values. a.u., arbitrary units; BOLD, blood oxygenation level dependent; FDR, false discovery rate; $i$PC$_{Multi}$, multivariate instantaneous phase coherence; TR, repetition time.

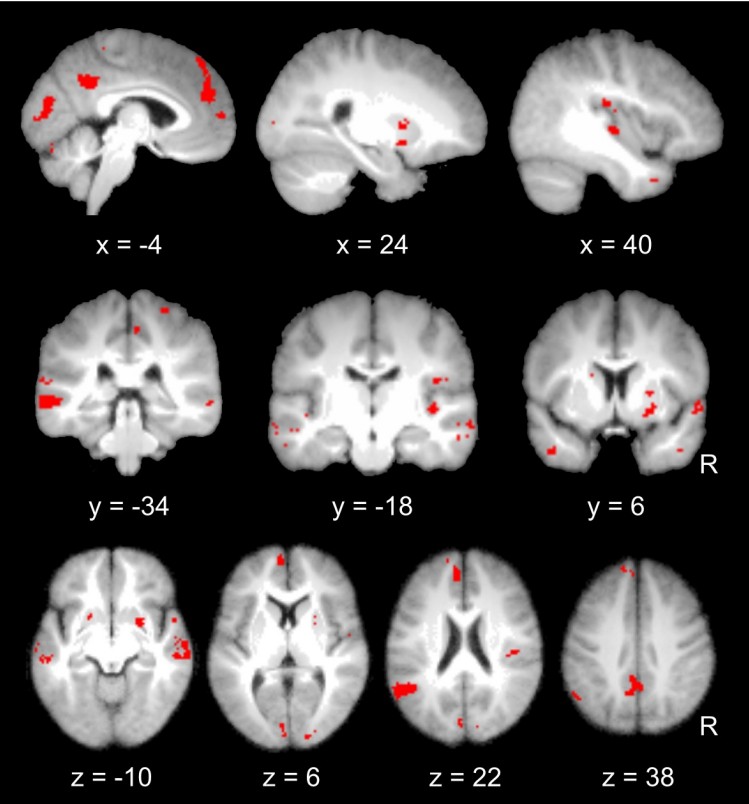

**Fig 7. Overlap between brain activation patterns associated with synchronization between emotion components estimated by the brain-based synchronization and the model-based synchronization indices.** Effects are presented on sagittal, coronal, and axial slices of a mean image created by averaging the participants' normalized structural images. Individual beta maps used for this figure are available on Neurovault at https://identifiers.org/neurovault.collection: 8740.

regressor in the first-level fMRI analyses of each participant, and entered the corresponding contrast into a second-level one-sample *t* test.

Remarkably, despite being derived from different measures, this model-based analysis revealed a set of cortical and subcortical sensory-motor regions with substantial overlap with our brain-based analysis, particularly in basal ganglia (caudate and putamen) and right posterior insula, but also primary sensory-motor cortices, PCC, medial PFC (mPFC), and antero-lateral temporal areas around the superior temporal sulcus (STS) (Fig 7, Fig D in S1 Fig; Table G in S1 Table). Areas showing the most consistent modulation by dynamic synchronization between emotion components were further highlighted by computing the overlap between brain maps associated with the 2 synchronization measures (Fig 7). These regions appear to constitute a "core" network that is uniquely engaged when brain activity representing distinct emotion components becomes highly synchronized with each other, based both on a data-driven analysis of fMRI time courses and on an independent computational model using peripheral measures during gameplay.

## Discussion

Despite being well established among psychological theories of emotion, appraisal-driven componential models have scarcely been explored in neuroscience. However, componential accounts neatly accord with modern views of brain functions in terms of multidimensional

neural networks with ongoing reciprocal dynamic interactions [21–23]. Additionally, most neuroimaging studies of emotion employed passive tasks that rely on the perception or recognition of emotions using indirect stimuli (e.g., faces), typically assigned to predefined emotion categories (e.g., fear), without first-person and self-relevant features inherent to emotional experience [but see 24, 25, 26].

We addressed these limitations by using an emotion-eliciting video game paradigm, combined with both model-based and data-driven analysis of fMRI that incorporated theory-driven parameters. Video games provide the opportunity not only to generate an interactive environment during which participants are actively engaged in a task evoking various emotions but also to systematically control situational parameters and appraisal processes postulated by psychology theories [20, 27, 28]. By promoting active engagement and different approach-avoidance situations, video games also allow measuring action tendencies and eliciting transient expressive and physiological reactions. As such, video games constitute an effective and flexible tool in the study of emotions and their componential features, in particular under the theoretical assumptions of the CPM. Here, our participants played an arcade game manipulating 2 major Appraisals implicated in emotional responding (Goal Conduciveness and Coping Potential) while we monitored brain activity together with behavioral tendencies, physiological signals, and facial EMG. This allowed us to delineate distinct brain networks (and their activity time courses) associated with Appraisal, Motivation, Physiology, and Expression processes, the 4 nonexperiential components of emotion defined by the CPM [10].

Our Appraisal manipulations not only successfully influenced behavior and subjective emotional experience (as shown by both subjective ratings and game performance measures) but also engaged specific brain networks. Coping Potential recruited areas involved in action planning and uncertainty, whereas Goal Conduciveness modulated areas involved in value/reward processing and attention. Furthermore, as predicted, interaction effects between these 2 Appraisals evoked distinctive activations in limbic areas typically associated with emotion, including the amygdala, ACC, and anterior insula. These areas also respond to salience and relevance detection [29, 30], highlighting that brain activation patterns tend to overlap across various affective and nonaffective conditions [5], and speaking against strictly modular accounts of emotions. Although neither relevance nor salience was explicitly manipulated, both are inherently embedded in appraisals of Goal Conduciveness and Coping Potential. Indeed, relevance is often considered as a primary step that triggers emotion-eliciting appraisals [9]. Our behavioral and neuroimaging results thus converge to support a key role for appraisal processes and their interactions in the generation and differentiation of emotional responses.

Brain networks mediating Motivation processes were also found to recruit widespread regions, with approach and avoidance tendencies during the game implicating the OFC, ACC, and PCC, as well as right dorsolateral PFC, bilateral insula, and parietal areas. Similar regions have likewise been related to approach and avoidance in fMRI studies of goal-directed behaviors [31, 32], further demonstrating the validity and efficacy of our paradigm.

Finally, we found that measures of spontaneous facial expressions during the game covaried with a motor network comprising the primary motor cortex, cerebellum, and striatum, consistent with circuits controlling facial muscles [33, 34], whereas peripheral autonomic physiology was associated with activity in subcortical areas of the thalamus, basal ganglia, amygdala, as well as the posterior insula and right PFC, again in line with previous studies focusing on affective modulation of peripheral physiology [35, 36].

Overall, these results indicate that each of the Appraisal, Motivation, Expression, and Physiology components rely on distributed patterns of subcortical and cortical activity, with partial

overlap across components, in accord with current network-based accounts of affective and cognitive functions.

More critically, a key assumption of the CPM is that emotions (with their associated feelings) consist of transient episodes emerging from synchronized changes across component processes. To identify brain regions involved in this dynamic coordination, we derived 2 independent measures of synchronization between components, first using empirically derived time courses of brain network activity, and second using a computational model fed with behavioral and peripheral measures. Brain-based and model-based approaches converged to unveil neural representations of component synchronization in a bilateral sensory-motor network centered on basal ganglia, posterior insula, and somatosensory/association areas in parietal cortices, as well as midline areas in the dmPFC and PCC (Fig 7, Fig D in S1 Fig; Table G in S1 Table).

Such involvement of somatosensing and motor areas during component synchronization is consistent with a conceptual framework considering emotion as an embodied action preparation mechanism that sets up the organism to promote adaptive responses to relevant events [37, 38]. The basal ganglia may constitute a key part within this synchronization network, as they provide a unique site of convergence between multiple cortical regions traditionally with specialized affective, cognitive, or sensory-motor processes [39, 40], and allows for a precise moment-to-moment coordination and patterning of neural activity across distant areas [41]. Indeed, apart from their participation in motor preparation and goal-directed action [42], the basal ganglia may play a critical role in integrating functional information from different component systems in order to generate emotion-specific patterns in action and cognition [43].

Another key part of the synchronization network was the right posterior insula, a region that also receives convergent multimodal information about bodily states [17, 44] and was previously highlighted as a crucial site for mapping ongoing changes in somatic functions and subcortical pathways activated during emotional responses [45, 46]. It is tempting to speculate that, whereas synchronization effects in basal ganglia reflect mechanisms necessary to orchestrate appropriate behavioral changes across brain-wide networks in emotion-eliciting contexts, synchronization in posterior insula might result from monitoring interoceptive processes through which these changes generate the subjective bodily experience (or feeling component) of emotions [47, 48]. Of note, unisensory (e.g., visual and auditory) and multisensory (e.g., STS) cortical regions were also modulated by synchronization between components, further reflecting that emotion episodes imply high functional coupling between multiple systems in the organism, including perceptual pathways engaged by external stimuli [10, 49], and possibly representing the affective value of their sensory attributes [50].

In addition, component synchronization engaged mPFC and PCC, 2 areas commonly implicated in introspective processing and access to self-related information in memory [51]. Recent fMRI studies reported that these midline regions selectively activate during conscious attention to self-experienced emotions (as opposed to the sensory content of experience) [52] and hold segregated representations of both basic and non-basic emotion categories in voxel-wise patterns identifiable with machine-learning classification analysis [53, 54]. Moreover, both the mPFC and PCC respond to emotional categories and valence in supramodal codes that generalize not only across different perceptual inputs from facial and vocal expressions [55] but also to self-experienced emotional events [56, 57]. These regions may be well placed to integrate sensorimotor and visceral interoceptive information organized in basal ganglia and insula centered circuits, respectively, together with ongoing cognitive and memory processes, in order to form higher-order representations of current bodily and mental states of the self that are inherent to conscious affective experience [58].

The notion that transient episodes of coordinated components and embodied representations underlie emotion is not unique to the CPM but is also present in other models to some extent [38, 59, 60]. For example, in the somatic marker hypothesis proposed by Damasio and colleagues [46], emotions result from an integration of concomitant changes in the internal milieu (Physiology), musculoskeletal system (Expression), and actions or decisions (Motivation). However, according to Frijda [61], a characterization of emotion as felt representations of body states is insufficient because it lacks an intentional content. Instead, action readiness (implying a directed relationship with the environment) is a more fundamental ingredient defining emotions and their specificity [37]. Our findings may help reconcile and go beyond these proposals by showing that the synchronization of large-scale brain networks mediating different emotion components selectively recruits neural systems with a key role in integrating sensory-motor signals and self-relevant information to orchestrate goal-directed behaviors.

This fMRI study directly addressed the multi-componential nature of emotions and tested their dynamic interplay at the brain network level with a first-person gameplay and computational modelling based on theory-driven parameters. In doing so, our work provides new insights into the functional organization of human emotions. However, our study is not without limitations. First, the Goal Conduciveness Appraisal was manipulated through gains or losses associated with the different monsters, possibly confounded with reward mechanisms. Effects of Goal Conduciveness were not explained by classic reward-responses alone, because comparing conditions associated with different rewards across different levels of the same Appraisal condition did not exhibit comparable brain patterns (Fig E in S1 Fig, Table H in S1 Table). However, other paradigms would be useful to disentangle goal conduciveness/obstruction from reward/punishment.

Second, we chose 2 major Appraisals with well-established roles in emotion elicitation, but interactive game paradigms could also investigate the neural basis of other appraisals (e.g., novelty, social norms). Furthermore, our study did not elucidate the precise cognitive mechanisms of particular appraisals or their neuroanatomical substrates but rather sought to dissect distinct brain networks underlying appraisals and other emotion components in order to assess any transient synchronization among them during emotion-eliciting situations. Importantly, even though different appraisals would obviously engage different brain networks, a critical assumption of the CPM is that synchronization between these networks and other components would arise through similar mechanisms as found here.

Third, our task design and event durations were chosen for fMRI settings, with blocked conditions and sufficient repetitions of similar trials. The limited temporal resolution of fMRI did not allow the investigation of faster, within-level dynamics which may be relevant to emotions. Additionally, this slow temporal resolution and our brain-based synchronization approach are insufficient to uncover fast and recurrent interactions among component networks during synchronization, as hypothesized by the CPM. Nonetheless, our computational model for the peripheral synchronization index did include recurrence as one of its parameters, allowing us refine our model-based analysis of network synchronization in ways explicitly taking recurrent effects into account (see S1 Text and Table J in S1 Table). In any case, neither the correlation of a model-based peripheral index nor an instantaneous phase synchronization approach could fully verify this hypothesis at the neuronal level using fMRI. To address these limitations, future studies might employ other paradigms with different game events or other imaging analyses and methodologies with higher temporal resolution. Higher temporal resolution may also help shed light on causality factors hypothesized by the CPM, which could not be addressed here. Finally, our study focused on the 4 nonexperiential components of emotion, with feelings measured purely retrospectively for manipulation-check purposes. This approach was motivated conceptually by the point of view that an emotion can be characterized

comprehensively by the combination of its nonexperiential parts [10] and methodologically by the choice to avoid self-report biases and dual task conditions in our experimental setting. However, future work will be needed to link precise moments of component synchronization more directly to concurrent measures along relevant emotion dimensions, without task biases, as previously examined in purely behavioral research [20].

Nevertheless, by investigating emotions from a dynamic multi-componential perspective with interactive situations and model-based parameters, our study demonstrates the feasibility of a new approach to emotion research. We provide important new insights into the neural underpinnings of emotions in the human brain that support theoretical accounts of emotions as transient states emerging from embodied and action-oriented processes which govern adaptive responses to the environment. By linking transient synchronization between emotion components to specific brain hubs in basal ganglia, insula, and midline cortical areas that integrate sensorimotor, interoceptive, and self-relevant representations, respectively, our results provide a new cornerstone to bridge neuroscience with psychological and developmental frameworks in which affective functions emerge from a multilevel integration of both physical/bodily and psychological/cognitive processes [62].

## Methods

### Participants

Twenty-six right-handed participants with no history of neurological or psychological illness were included in the analyses (14 male; mean age: 23.81 y; SD: 4.71). Their Edinburgh Handedness Inventory score [63] was 74 ± 19.1 (mean ± SD) and Beck Depression Inventory score [64, 65] 4.96 ± 5.64 (mean ± SD; all scores < 30). Three additional participants were excluded from analyses, respectively, because of left-handedness, drowsiness during scanning, and excessive movement that prevented reliable physiological measurements. Participants had normal or corrected-to-normal vision. All gave written informed consent. The study was run in accordance with the Declaration of Helsinki and approved by the Research Ethics Committee of the Geneva University Hospital (CER 09–316 and BASEC 2018–02006).

All participants filled in several questionnaires that included an in-house assessment of their video game habits, as well as the SPSR questionnaire [66, 67], the BIS/BAS personality scales [68, 69], and the State-Trait Anxiety Inventories (STAI) [70]. A questionnaire about their game experience during our video game [71, 72] was also filled after the fMRI session, showing that participants felt relatively competent in playing the game and were positively engaged during the task (Table I in S1 Table: competence, flow, and affect facets).

### Experimental design and task

Participants completed the video game in which they were represented as a yellow agent who navigated different mazes (moving every 8 frames, i.e., at 7.5 Hz; see Fig 2) across different levels (or trials), with the goal of collecting as many points as possible and then reaching a final target location. Points could be obtained by picking coins up along the avatar's way (5 points each). At the beginning of each trial, 12 coins were displayed and distributed throughout the maze. Once those were picked up, additional coins would appear one by one at random times and random places.

This design allowed us to manipulate different Appraisal conditions across different levels. To vary Goal Conduciveness, on each level, the player was accompanied by one monster that also navigated in the maze and exhibited one of 3 possible behaviors (in different trials). These behaviors were signaled by the monster's color and shape, informing about its movement ability and consequences of touching it (i.e., gain or loss of points). In neutral monster conditions,

the monster moved randomly and touching it had no consequences for the participant. In good monster conditions, the monster chased the player with 0.85 probability and moved randomly otherwise. Touching this type of monster yielded 10 points. Good and neutral monsters moved with the same speed as the player. In contrast, bad monsters moved faster (every 7 frames, i.e., at 8.6 Hz) and continuously chased the player. Touching a bad monster made the player lose 100 pts.

To manipulate Coping Potential, on half the trials, participants were given the possibility to activate a super-power. The super-power option was signaled by a small "magic potion" icon blinking on top of the yellow avatar. When activating power (by pressing a dedicated key at any time during the trial), the avatar changed its color from yellow to orange, and touching monsters led to different outcomes. Good monsters now yielded 100 points, while bad monsters mitigated the loss to 10 points. Consequences of touching neutral monsters remained unaltered (baseline condition). Once the super-power was activated, the avatar remained in this mode until the end of the level.

Together, this resulted in a 2 × 3 design with factors of (i) Coping Potential (no-power, power) and (ii) Goal Conduciveness (good, neutral, and bad monsters) (Fig 2A). These 2 factors were manipulated across levels, according to a standard block design (each navigation period in the maze corresponded to one block). Please note that different combinations of Appraisals generate different emotion types (e.g., low power with bad monsters should elicit anxiety, while high power and good ghost should elicit satisfaction), but emotion categories were not prespecified by design.

To proceed to the next trial/level, participants had to move their avatar to a teleporter at the top of the maze (Fig 2B). This teleporter was placed behind a closed door that opened automatically after a certain time. To avoid having participants navigating in the maze indefinitely, a countdown period was introduced that set a time delay (4 s for neutral, 6 s for good and bad monsters) during which participants had to reach the teleporter once the door was opened. If they did not reach the teleporter within the allotted time ("too late" trials), all points gathered during that level were lost. The countdown period was signaled by an audiovisual cue. Each pre-countdown block lasted 8 s, ensuring the same amount of time for each experimental condition, while the countdown itself varied but was modeled separately. Reaching the teleporter was followed by a brief interval (1.5 s) before the next level.

Participants played 3 runs inside the MRI scanner, each comprising 72 levels and lasting approximately 15 min. This amounted to 12 blocks per condition per run and 36 blocks per condition in total. The order of conditions was pseudo-randomized such that all possible transitions between different conditions took place approximately the same number of times. To ensure that participants were able to navigate properly in the maze and to minimize potential practice effects during fMRI, participants took part in a training session outside the scanner (see below).

Finally, to move the avatar in the maze, participants only needed to press a key when wanting to change direction or after they were stopped by an encounter with a monster, which minimized the number of necessary keypresses.

## Training session

To ensure that participants were able to navigate properly in the game maze and minimize learning effects during fMRI, a training session was first given outside the scanner. This training session took place on the same week as the fMRI session, with an average interval of 4.28 ± 0.84 (± SD) d. During this session, participants completed one run of a training game that familiarized them with the key layout and game display without including any Appraisal

manipulation. Briefly, the points gained during this training phase were restricted to collecting the coins (10 points each), which appeared one by one at random locations and for only a short period of time, thereby forcing participants to navigate through the entire maze. To practice the use of the super-power, the coin value was multiplied by 5 if it was picked up when the super-power option was presented and activated. Finally, to familiarize participants with the presence of monsters, a neutral monster was present in every level, and participants were told that this monster had no effect. The same general display was used as in subsequent game sessions. A second training run was completed in case participants did not feel completely at ease with the key display and/or navigation after the first run (3 participants).

After this initial familiarization period, participants completed a short version of the main game (2 levels per condition), followed by a final full run comprising the same number of levels (12 per condition) as the experimental scanner session. The short version of the main game was again played at the beginning of the scanner session, to ensure that the participants got accustomed to playing in the new environment.

During the training session, we also tried to mimic as much as possible the keyboard conditions used inside the scanner by adapting a numeric pad with buttons placed at approximately the same distance and layout of the keypad used inside the scanner (see "Stimuli presentation") and asking participants to hold the numeric pad on their leg and below the table, with buttons out of their view. This forced them to rely on manual feedback only to navigate properly inside the maze.

## Visual stimuli

Monster and player avatars were designed using Inkscape (https://inkscape.org/). All 3 monsters had exactly the same (neutral) facial expression, differing only in shape and color, which signaled the monster type. Neutral monsters were always gray, while good and bad monsters were pseudo-randomly counterbalanced between blue and purple across participants (see Fig 2A). Six different maze layouts were used, counterbalanced across conditions. From level to level, mazes could have one of 5 different colors that were also pseudo-randomly attributed.

Countdown periods were signaled visually by displaying a numeric countdown panel below the score panel, which in turn showed the current total score in the run. In the second and third runs, the score panel also included the best score from previous runs in order to maintain motivation across all game runs. To avoid confounds in the fMRI data associated with systematic eye movements towards only one side of the visual display, the location (i.e., to the left or to right of the play maze) of the countdown and score panels was counterbalanced across participants. A table showing monster labels and values for the different conditions was displayed on the opposite side of the maze (see Fig 2B).

As mentioned in "Experimental design and task," movements of the monsters changed according to the current Goal Conduciveness condition. The movement was controlled using the A* algorithm [73], which is a pathfinding algorithm that aims at finding the smallest cost (here shortest distance) path between prespecified start and target nodes (here corresponding to positions in the maze). The monster's path was updated at every frame, with the start node always being its current position, while the target node changed according to the monster type. When the monster was neutral, the target node was chosen at random from one of the maze positions that were at a safety distance away from the current player's position. When the monster was bad, the target node was always the current player's position. When the monster was good, the target node was the player's position in 85% of the times and a random position otherwise.

## Auditory stimuli

Depending on the monster and power context, different outcomes resulted from touching a monster, and each outcome was associated with a specific sound. To avoid confounds associated with the processing of completely different sounds coupled with our experimental conditions, 5 different sounds having similar low-level characteristics were created (during the neutral monster condition the outcome was the same regardless of the power situation), by applying different frequency modulations to the same amplitude modulated carrier sound. Specifically, the signals were of the form:

$$s = (1 + depth_{am} * \sin(2\pi f_{am} t)) * (amp * \sin(2\pi f_{carrier} t) + (-index_{fm} * \sin(2\pi f_{fm} t))) \qquad (1)$$

where the frequency of the carrier sound equaled $f_{carrier}$ = 440 Hz and its amplitude $amp$ = 0.5. The amplitude modulation used a frequency $f_{am}$ = 40 Hz and a modulation depth $depth_{am}$ = 0.6. Hence, the signals only differed in their frequency modulation rate ($f_{fm}$), which changed between 1 and 5 Hz in increments of 1 Hz, while a constant modulation index $index_{fm} = \frac{max\_freq\_change}{f_{fm}} = 100$ was used. These signals were all 500 ms in duration and were multiplied with a Hann window to avoid clicking effects in the beginning and end of the sounds. To further minimize possible confounds, the sounds resulting from encounters with different monster types were counterbalanced across participants.

Additionally, another distinct sound was associated with opening of the door to the teleporter, taken from a publicly available library (http://soundbible.com/1357-Metal-Latch.html), while the countdown was signaled by spoken numbers (http://soundbible.com/2008-0-9-Male-Vocalized.html). Other sounds associated with picking up coins, reaching the teleporter, and activating the super-power mode were custom made using Bfxr software (http://www.bfxr.net/).

## Stimuli presentation

Visual and auditory stimuli were presented using Psychophysics Toolbox version 3.0.13 [74, 75] running on MATLAB 2015b (MathWorks, Natick, MA) and a 64-bit Windows 7 operating system (Microsoft, Redmond, WA). The game was programmed using the object-oriented programming capabilities of MATLAB.

All visual stimuli were displayed on a 23" LCD monitor (Cambridge Research Systems Ltd, Kent, UK; model: BOLDscreen 23; resolution: 1,920 × 1,080 pixels, dimensions: 50.9 cm × 29 cm, refresh rate: 60 Hz, viewing distance: approximately 125 cm), seen by the participant through a mirror mounted on the MR-head coil.

Auditory stimuli were heard via type HP AT01 earphones composed of an electro-dynamic earphone driver and 46-cm-long air tubes (Cambridge Research Systems Ltd, Kent, UK), connected to standard-sized insert earphones (Canal Tips, Comply, MN, US). The earphones were connected to the stimulus computer via a MR Confon amplifier unit (MR Confon GmbH, Magdeburg, Germany).

Participants navigated in the game using their right hand and an MR-compatible 5-Button Diamond Fiber Optic Response Pad (Current Designs Inc, Philadelphia, PA; model: HHSC-1x5-D), connected to the stimulus computer via a FIU-932-B electronic interface. Participants used the green, pink, yellow, and red buttons to navigate up, down, left, and right, respectively, while the blue button was reserved to activate the super-power.

## Appraisal and feeling ratings

Directly after the scanner session, participants performed an Appraisal check and reported their subjective feelings relative to each manipulated condition during a separate run outside the scanner. For each condition of our $2 \times 3$ design, participants saw screenshots of the game in which the player and monster avatars were placed randomly within the maze but always at a specific distance from each other. This distance was calculated in a participant-specific manner, based on the average distance between the 2 characters during actual gameplay. There were 4 repetitions per condition, yielding a total of 24 screenshots.

For each screenshot, participants were asked to answer (5-point Likert-scale ranging from "Strongly Disagree" to "Strongly Agree") the following questions about how they had generally appraised similar situations during the game:

1. For the Goal Conduciveness judgments: "Imagine you would now touch the monster. Do you think that the outcome of touching the monster is in line with the goal of the game?"

2. For the Coping Potential judgments: "Imagine you would now touch the monster. Do you think that the outcome of touching the monster has been modified by the player's actions?"

To probe subjective emotion feelings, participants also had to rate how they felt in the corresponding situation by answering the question "Generally, how did a level of this type made you feel?" along a range of emotion categories (boredom, satisfaction, frustration, anxiety; 5-point Likert-scale ranging from "Not at all" to "Very much") and in terms of valence, arousal, and dominance dimensions (using a 5-level self-assessment manikin scale). For each participant, the modal value across the 4 repetitions of each condition was then calculated and used to compute the frequency distribution of different responses across participants (for each question and each condition separately).

## Motivation component

In line with the CPM proposed by Scherer [10] and action readiness theories proposed by Frijda [6, 76], we defined the Motivation component in terms of action tendencies such as approach and avoidance, which we could measure due to the interactive nature of our task. Behavioral measures taken to represent these tendencies were (i) the average number of times the player was caught from the back (ApproachTail; avatar facing away from the monster when caught; representing avoidance) or (ii) from the front (ApproachHead; avatar facing towards the monster when caught; representing approach); (iii) the time spent during countdown periods (CountdownTime; the longer, the higher the potential to win or lose extra points with good or bad monsters, respectively); and (iv) the number of coins collected in a level (EatenCoins; representing a change in the task focus from the monster to the coin in order to gain points). Because the neutral condition was introduced as a baseline condition and the above indices did not change across the Coping Potential manipulation for this condition (see Results), for each of the indices above we subtracted the corresponding neutral condition value from those of the good and bad conditions and entered them separately in a 2 (Coping Potential: no-power, power) $\times$ 2 (Goal Conduciveness: good versus neutral, bad vs neutral) repeated measures analysis of variance. To provide a complete overview of the data, we report the raw (un-subtracted) average values (±SD) for each of the indices in Table 1.

To be used in fMRI analyses, each motivation index was converted into a parametric modulator regressor using SPM functions modulating all pre-countdown periods (i.e., first 8 s of each level), after collapsing the onsets of levels across experimental conditions, demeaning, and convolving the resulting time course with the HRF.

These indices were equally converted into time courses to be used in the calculation of the peripheral synchronization index between emotion components (Peripheral Synchronization Index). Specifically, EatenCoins, ApproachHead, and ApproachTail indices were converted into binary event time courses, in which the value was set to 1 at event time points and 0 otherwise. Finally, the CountdownTime time course increased incrementally with the number of seconds elapsed on the countdown clock during countdown periods, and was 0 otherwise. These time courses were all up-sampled to 120 Hz.

## Physiology component

The physiology component was measured by recording heart rate and RR as well as skin conductance (or EDA). All measures were acquired using an MP150 BIOPAC acquisition system coupled with the Acqknowledge Software (BIOPAC Systems, Goleta, CA; Acqknowledge version 4.2 for PC/Windows). Sensors were connected to MR-compatible MECMRI-1 cables (BIOPAC Systems), routed through a patch panel to amplifiers located outside the scanning room. The acquisition sampling rate equaled 5 kHz. Cardiac rhythm and respiration were recorded using a channel sampling rate of 625 Hz.

Cardiac rhythm was measured using a TSD2000-MRI plethysmograph attached on the distal phalanx of the ring finger of the left hand and connected to a PPG100C-MRI amplifier (gain: 100, low-pass filter: 10 Hz, high-pass filter: 0.5 Hz). Cardiac time courses were down-sampled to 120 Hz and band-pass filtered using lower and upper cutoff frequencies of 1 Hz and 40 Hz, respectively. These preprocessed time courses were fed into custom-written Matlab scripts to identify the peaks associated with pulse beats, which were manually verified and thereafter used to compute the instantaneous (i.e., beat to beat) heart rate.

Respiration was measured with a TSD221-MRI respiratory belt around the base of the rib cage, connected to a RSP100C-MRI amplifier (gain: 10, low-pass filter: 10 Hz, no high-pass filter). Respiration time courses were equally down-sampled to 120 Hz and band-pass filtered using lower and upper cutoff frequencies of 0.05 Hz and 1 Hz, respectively. The RR was then calculated analogously to the heart rate.

Skin conductance was measured by placing MR-compatible Ag/AgCl ClearTrace[2] snap electrodes (ConMed Corp., NY) on the distal phalanges of index and middle fingers of the left hand, which were connected via LEAD108C leads to a EDA100C-MRI amplifier (gain: 5, low-pass filter: 10 Hz, no high-pass filter). Skin conductance time courses were down-sampled to 120 Hz and low-pass filtered with a cutoff of 1 Hz. Time courses were subsequently corrected for movement artefacts and signal dropouts using custom-written Matlab scripts. The corrected time courses were then separated into tonic and phasic components using the continuous decomposition analysis option from the Ledalab toolbox [77] with an optimization factor of 4 and the default values for the remaining parameters. Participants were instructed to keep their left hand still to avoid movement artifacts in the electrodermal and cardiac recordings.

Synchronization of these physiological measures with the experimental protocol was achieved through the simultaneous acquisition of digital input markers that were sent from the experimental PC to the BIOPAC system via an 8-bit parallel port. Unless stated otherwise, based on the optimization criterion for the coefficients of FIR filters used in the Acqknowledge software (see Acqknowledge 4 Software Guide), the high-, low-, and bandpass filters used a Blackman window with order N = 4*Fs/Fc.

For fMRI regression analysis, the RR time courses were resampled to the MR volume onsets (i.e., to the TR = 600 ms, see "fMRI data acquisition") and convolved with the HRF. The phasic skin conductance time courses were simply resampled to the MR volume onsets but were not

convolved with the HRF, as this signal is comparable in shape and latency to the BOLD hemo-dynamic response [78].

## Expression component

EMG activity of facial muscles was recorded by placing 2 Micro NeoLead IEC radiolucent electrodes (Philips Medical Systems, Andover, MA) over the left corrugator supercilii and the zygomaticus major muscles, respectively, according to guidelines recommended by Fridlund and Cacioppo [79] after appropriate cleaning of the skin. An additional electrode was placed on the forehead, medially above the inion, to serve as the ground reference. The data were acquired using a BrainAmp MR system (Brain Products GmbH, Gilching, Germany). The acquisition sampling rate equaled 5 kHz. Electrodes were connected via a 16-channel ExG AUX input box to a BrainAmp ExG MR amplifier (low-pass filter: 1 kHz, high-pass filter: 10 Hz, resolution: 10 μV), placed together with the PowerPack at the head end of the scanner bore and operated using the BrainVision Recorder software. In order to facilitate MR artefact correction of the EMG signals, MR volume triggers were recorded using the SyncBox system to synchronize the sampling rate of the amplifier with the scanner clock system. Synchronization with the experimental protocol was achieved through simultaneous recording of digital input markers, sent from the experimental PC to the BrainAmp system via an 8-bit parallel port.

To confirm the validity of signals from each muscle, EMG tests were performed in the scanner prior to the experimental runs, by asking participants to frown and smile when prompted. Onset times of each prompt were saved through the simultaneous recording digital input markers.

EMG data were high-pass filtered at 20 Hz to remove slow signal fluctuations and movement artefacts. They were then passed on to the FMRIB plug-in for EEGLAB (provided by University of Oxford Centre for Functional MRI of the Brain) [80, 81] using the adaptive noise cancellation option with a low-pass filter of 400 Hz to correct for MR-related artifacts (average window size = 21). The instantaneous amplitude of the corrected signals was computed by taking the absolute value of the Hilbert transform for each of the 2 muscles separately, followed by averaging these values across the 2 muscles.

For fMRI regression analysis, to obtain one EMG value per MR volume (i.e., per TR = 600 ms, see "fMRI data acquisition"), we calculated the average instantaneous amplitude per volume and normalized the resulting signal by the average amplitude across an entire experimental run. Finally, this signal was convolved with the HRF. For computation of the peripheral synchronization index (see "Model-based synchronization index derived from peripheral measures"), the unconvolved signals were further up-sampled to 120 Hz.

## Model-based synchronization index derived from peripheral measures

According to the CPM, an emotional episode and its associated feeling component emerge once a certain degree of synchronization between the constitutive emotion components is attained [10]. Meuleman and colleagues recently developed a computational model (Emergent Liquid State Affect [ELSA]) that calculates a continuous statistical index reflecting such synchronization over time [20]. Briefly, the model applies a combination of wavelet transforms, liquid state machines, and penalized regressions to derive the temporal synchronization index based on the time course of each component and their modelled interactions. To do so, the model fits different sub-models for the Motivation, Physiology, and Expression activity in parallel, whereby each component is included as an input variable in the models of other components. As the Appraisal component is manipulated by experimental conditions, it is only

included as an input variable in the models of other components and never considered as a response variable itself. In each of the sub-models, wavelet transforms are applied to the input variables to account for different temporal scales and unfolding of the signals, which are then entered, along with interactions between different components, into a liquid state machine. Liquid state machines, also called echo state machines or reservoir computing, are large recurrent neural networks (RNN) that allow for nonlinear effects and feedback connections, thereby effectively constituting a dynamic system [82, 83]. Specifically, a liquid state machine decomposes an inputted time-series ($x_t$) inside a massive, randomly connected RNN and then uses the decomposition for further modelling rather than the original inputs. Setting up the RNN required defining an initial liquid size (number of units in the RNN, L = 1,500), a reduced eigenliquid size (EL = 100), the input-to-liquid ($W^{IN}$ = 0.01) weights, as well as a leaky integration factor ($\alpha$ = 0.01), which controls the amount of exponential decay of past states. Liquid-to-liquid unit weights ($W_{\lambda max}$) were set sparsely and randomly, and then scaled such that the leading eigenvalue of the weight matrix $\lambda$ is smaller than a maximal limit ($\lambda_{max}$ = 0.8), with longer fading memory for smaller limits of $\lambda$ (please refer to [82] for a full description of the liquid state machines and corresponding parameters). In sum, the updating equations for the liquid at each time $t$, can be calculated by:

$$Ł_t = \tanh(W^{IN}[1; x_t] + W_{\lambda_{max}} L_{t-1}) \tag{2}$$

$$L_t = (1 - \alpha)L_{t-1} + \alpha Ł_t \tag{3}$$

To run the model, we divided the 3 runs of each participant in training (2 runs) and test data sets. The parameter values were selected based on a forward leave-one-run-out optimization search using pilot data. In addition, given that some parameters are randomly determined, we ran 20 repetitions per run using different seed initializations each time, and used the averaged results across repetitions in further analyses.

The output of the liquid state machine was reduced to a smaller eigenliquid by principal component analysis, whose component scores were then used to predict the response time-series of the specific sub-model using penalized regression. Finally, the outputs of sub-models were used to compute the temporal synchronization between components. This was achieved by computing the covariance matrix between the different time courses within 8-s sliding windows (using steps of one sample) and subsequently calculating the spectral radius as the maximum eigenvalue of the covariance matrix normalized by the sum of all eigenvalues. This spectral radius time course constituted the temporal synchronization index, whose values ranged from 0 to 1, whereby a value of 1 meant that the components were highly synchronized at that specific time point. This index was finally entered as a regressor in our fMRI analysis (see below).

For a more detailed description of liquid state machines and the complete ELSA steps, see Meuleman 2015 [20].

## fMRI data acquisition

A 3T TIM Trio System (Siemens, Erlangen, Germany) was used to acquire both high-resolution structural images (MPRAGE, TR = 1,900 ms, TE = 2.27 ms, TI = 900 ms, flip angle = 9˚, FOV = 256 × 256 mm$^2$, image matrix 256 × 256, 192 sagittal slices, voxel size = 1 mm isotropic, 32-channel head coil) and T2$^*$-weighted axial echoplanar images (EPIs) with BOLD contrast (GE-EPI, TR = 600 ms, TE = 32 ms, flip angle = 52˚, FOV = 210 × 210 mm$^2$, image matrix 84 × 84, 48 axial slices, slice thickness = 2.5mm, with a multiband acceleration factor of 6, voxel size = 2.5 mm isotropic, 32-channel head coil). B$_0$ field maps (GR, 2D, TR = 528 ms, short

TE = 5.19 ms, long TE = 7.65 ms, flip angle = 60˚, FOV = 210 × 210 mm$^2$, image matrix 84 × 84, negative blip direction, slice thickness = 2.5 mm, 32-channel head coil) were also acquired to correct for static magnetic field inhomogeneities in the EPI images.

Each participant took part in a total of 3 experimental runs. On average, 1,490 (±49 SD) volume images were acquired for each run. Functional and field map images were acquired using the same field of view, with the matrix's z direction placed axial and co-planar relative to the anterior commissure-posterior commissure line. Whole-brain coverage was attained for all cortical and subcortical brain areas, with partial inclusion of the cerebellum and brainstem.

## fMRI data preprocessing

The fMRI data were preprocessed using SPM12 (Wellcome Department of Imaging Neuroscience, London; www.fil.ion.ucl.ac.uk/spm) [84]. Scans from each participant were realigned using the first as a reference, corrected for B$_0$ field inhomogeneities using phase maps obtained with the SPM12 FieldMap toolbox and co-registered to participants' anatomical images. The images were spatially normalized into MNI space using the parameters obtained from segmentation of the anatomical images, resampled to a spatial resolution of 2 × 2 × 2 mm$^3$ and spatially smoothed with a Gaussian kernel of 8 mm full-width at half-maximum. The time-series of all voxels were high-pass filtered to 1/128 Hz and pre-whitened using the "FAST" option from SPM12, which is based on exponential covariance functions and better suited for data acquired with short repetition times. The first 5 volumes were discarded to allow for T1-equilibration effects.

## fMRI data analyses

**Appraisal networks—Standard GLM approach.** At the subject level, the fMRI paradigm was modeled as a block design. Each run included pre-countdown periods of 8-s blocks for each experimental condition separately (2 Coping Potential × 3 Goal Conduciveness), with countdown blocks as an additional regressor of no interest shared across conditions with durations corresponding to the actual countdown time for each specific level. In addition, the onset of caught moments (i.e., onsets of when the monsters touched the player), auditory inputs (i.e., any type of sound, such as caught moments, countdown sounds, power up, etc.), and key-presses were modelled as events with separate regressors across all levels/conditions. All regressors were convolved with the canonical HRF and entered into the design matrix. Nuisance covariates included averaged cerebrospinal fluid and white matter time courses realignment parameters and their derivatives to account for residual motion artifacts, as well as respiration volume per time [85, 86], heart rate convolved with the cardiac response function [87], and RETROICOR [88] regressors (using third- and fourth-order Fourier expansions for the cardiac and respiration corrections, respectively), which allowed us to deal with physiological artifacts.

For each participant, condition-specific effects were then estimated according to the GLM by creating contrast images of each condition. To allow for random-effects analyses and inferences at the population level [89], contrast images were entered in a second-level RM-ANOVA that modelled the 2 factors experimentally manipulated in our game (Goal Conduciveness and Coping Potential Appraisals) and their interactions, as well as the subject factor to account for the repeated-measures design.

At the second level, we evaluated the following statistical comparisons to delineate brain networks differentially engaged by Appraisal processes. Effects of Coping Potential were identified by comparing power and no-power conditions pooled (i.e., summed) across Goal Conduciveness conditions. Effects of Goal Conduciveness were identified by comparing "good

(-neutral)" and "bad (-neutral)" conditions pooled (i.e., summed) across Coping Potential conditions (i.e., $good_{(power + no-power)}$ and $bad_{(power + no-power)}$). We also directly compared good and bad conditions (i.e., $good_{(power + no-power)}$ versus $bad_{(power + no-power)}$). In addition, we tested for interaction effects between these 2 factors with the contrasts "$bad_{(no-power > power)} > good_{(no-power > power)}$" and "$good_{(no-power > power)} > bad_{(no-power > power)}$."

**Emotion component networks and their representative time courses.** We identified the emotion component networks and their representative time courses through the following steps.

First, we restricted the fMRI data to gray matter (GM) voxels that were acquired for all participants, which was guaranteed by considering only voxels that were in the intersection between the GM mask averaged across participants and the second-level mask obtained from the standard GLM approach detailed above. These fMRI data were corrected for nuisance variables (see "Appraisal networks—Standard GLM approach") and for movement artefacts based on Power and colleagues [90] and were subsequently high-pass filtered (cut-off at 1/128 Hz) before being entered into further analyses.

Second, we divided the 3 runs of each participant in training (2 runs) and test data sets. The training step consisted in learning a map that can predict emotion components from imaging data in a data-driven way. The features associated with each emotion component were the instantaneous amplitude of the EMG signal averaged across the 2 muscles for the Expression component (see "Expression component"), whereas the Motivation component was defined using parametric regressors associated with the ApproachHead, ApproachTail, Countdown-Time, and EatenCoins indices (see "Motivation component"). For the Physiology component, our analyses were restricted to skin conductance and RR time courses (see "Physiology component"), as the heart rate time course produced too noisy data. Finally, for the Appraisal component we used the block regressors of each experimental condition convolved with the HRF.

We predicted the maps associated with each of these component features from fMRI data using regression analyses (Fig 6, Fig B in S1 Fig). In order to avoid overfitting to the training set, we used the 800 leading components (Fig C in S1 Fig) resulting from singular value decomposition (SVD) of the training fMRI data and performed the regression analysis in this latent space (Fig B in S1 Fig). After back-projection to the original voxel space, the resulting estimated feature maps were thus combined (i.e., summed) to yield emotion component maps for the Expression, Physiology, and Motivation components (Fig B in S1 Fig). For the Appraisal component map, this combination consisted in the linear combination of feature estimates that corresponded to the interaction between the Coping Potential and Goal Conduciveness as it best captures emotion-eliciting moments according to the CPM.

Finally, the emotion component maps obtained using the training data set were applied to the test run by computing the scalar product between the 2 at each time point [19], which resulted in a representative time course of network activity per component per run (Fig 6, Fig B in S1 Fig). These time courses were band-pass filtered between 0.001 and 0.15 Hz to allow deriving the instantaneous phase of the signals using the Hilbert transform (Fig B in S1 Fig) and subsequently calculate the synchronization index between the 4 emotion component networks (see below).

The emotion component maps obtained for each participant at the training step were also used to compute group-level emotion component maps. For brain patterns associated with Expression, Physiology, and Motivation, the respective emotion component maps were averaged across runs and entered into second-level one-sample $t$ test (for the EMG data) or ANO-VAs (for physiology and motivation data) using SPM. In each of these analyses, we computed positive and negative one-tailed tests pooling over all indices of the same component (i.e., summing across skin conductance and respiration regressors for Physiology, and across ApproachHead, ApproachTail, CountdownTime, and EatenCoins regressors for Motivation).

Please note that the goal here was to identify networks associated with general aspects of each component during the gameplay, not networks modulated by one particular feature of a component (e.g., skin conductance versus respiration), hence the component networks were considered across component features.

**Synchronization between component networks from brain-based measures.**   Synchronization between the different components at the brain level was computed by using the instantaneous phase synchrony as a similarity metric between each pair of network time courses [91]. This metric allowed us to identify transient moments of dynamic component network synchronization. Yet, unlike the periphery synchronization index, it does not model putative interactive recurrence between those networks, which is an assumption hypothesized by the CPM. To obtain a multivariate index of synchronization across the 4 networks, we calculated the eigenvalues of the pairwise similarity matrix at each time point and computed the spectral radius as the largest absolute eigenvalue divided by the trace of the eigenvalue matrix. This yielded a normalized network synchronization time course per run (Fig 6, lower right panel; Fig B in S1 Fig). We then thresholded these time courses by taking the time points whose values were above the 95th percentile and used these time points to compute a z-score map across participants and runs, similarly to what is done in point detection or co-activation patterns analyses [92]. To restrict our maps to significant voxels, we converted these z-scores into $p$-values and thresholded the maps using an FDR [93] of q < 0.01 (Fig 6, lower left panel).

**Synchronization between component networks from peripheral model-based measures.**   To map brain activity patterns co-varying with the peripheral synchronization index, as derived from computational modelling (see above), we convolved this temporal index with the HRF and included it as an additional regressor in the first-level GLM model of each participant (see "Appraisal networks—Standard GLM approach"). The beta maps corresponding to this regressor (taken from each participant and summed across runs) were then entered into a second-level one-sample $t$ test.

**Statistical inference and thresholds.**   Due to the large extent of activation clusters when using a height threshold of $p < 0.001$ and a cluster level of $p_{FWE} < 0.05$ in the standard GLM approach, to allow for more reliable functional localization, our results for Appraisal manipulations are reported with a voxel-level threshold of $p_{FWE} < 0.05$. Unless stated otherwise, other activations are reported with a voxel height threshold of $p < 0.001$ and clusters at $p < 0.05$ corrected for multiple comparisons (family-wise error rate) based on Gaussian random field theory within a GM mask obtained from the brain segmentation images (averaged across participants). This GM mask was restricted to voxels acquired for every participant (i.e., group mask used in second-level SPM analyses) and eroded (using spm_erode.m) in order to minimize the inclusion of voxels from other tissues.

## Supporting information

**S1 Text. Supplementary methods.**
(DOCX)

**S1 Table. Supplementary tables.**
(DOCX)

**S1 Fig. Supplementary figures.**
(DOCX)

**S1 Data. Includes data for Fig 3.**
(7Z)

**S2 Data. Includes data for Table 1.**
(7Z)

**S3 Data. Includes data for S1C Fig.**
(7Z)

## Acknowledgments

We thank Gelareh Mohammadi, Michał Muszynski, and Klaus Scherer for helpful discussions and suggestions, as well as Maya Burckhardt for assistance during data acquisition. This study was conducted on the imaging platform at the Brain and Behavior Lab (BBL) and benefited from support of the BBL technical staff. Part of the computations was performed at University of Geneva on the Baobab cluster.

## Author Contributions

**Conceptualization:** Joana Leitão, Ben Meuleman, Patrik Vuilleumier.

**Data curation:** Joana Leitão.

**Formal analysis:** Joana Leitão.

**Funding acquisition:** Joana Leitão, Dimitri Van De Ville, Patrik Vuilleumier.

**Investigation:** Joana Leitão.

**Methodology:** Joana Leitão, Ben Meuleman, Dimitri Van De Ville, Patrik Vuilleumier.

**Resources:** Patrik Vuilleumier.

**Software:** Joana Leitão, Ben Meuleman.

**Supervision:** Patrik Vuilleumier.

**Visualization:** Joana Leitão, Ben Meuleman, Dimitri Van De Ville, Patrik Vuilleumier.

**Writing – original draft:** Joana Leitão, Patrik Vuilleumier.

**Writing – review & editing:** Joana Leitão, Ben Meuleman, Dimitri Van De Ville, Patrik Vuilleumier.

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
