## [Editor Report · Decision Letter 0]

29 Feb 2020

Dear Dr Leitao, 

Thank you for submitting your manuscript entitled "Brain networks of emotions in sync: computational imaging of video-game playing" for consideration as a Research Article by PLOS Biology.

Your manuscript has now been evaluated by the PLOS Biology editorial staff, as well as by an Academic Editor with relevant expertise, and I am writing to let you know that we would like to send your submission out for external peer review.

Please re-submit your manuscript within two working days, i.e. by Mar 04 2020 11:59PM.

Kind regards,

Gabriel Gasque, Ph.D.,

Senior Editor

PLOS Biology

---

## [Decision Letter · Decision Letter 1]

8 Apr 2020

Dear Dr Leitao,

Thank you very much for submitting your manuscript "Brain networks of emotions in sync: computational imaging of video-game playing" for consideration as a Research Article at PLOS Biology. Your manuscript has been evaluated by the PLOS Biology editors, by an Academic Editor with relevant expertise, and by two independent reviewers. A third reviewer had also agreed to send comments, but is now significantly delayed. Thus, we have decided to move forward with only two sets of comments, not to delay the decision any longer. If belatedly the third reviewer submits the review, I'll forward it to you.

In light of the reviews (below), we will not be able to accept the current version of the manuscript, but we would welcome re-submission of a much-revised version that takes into account the reviewers' comments. We cannot make any decision about publication until we have seen the revised manuscript and your response to the reviewers' comments. Your revised manuscript is also likely to be sent for further evaluation by the reviewers.

We expect to receive your revised manuscript within 2 months. 

**IMPORTANT - SUBMITTING YOUR REVISION**

*Re-submission Checklist*

*Published Peer Review*

*PLOS Data Policy*

*Blot and Gel Data Policy*

Sincerely,

Gabriel Gasque, Ph.D., 

Senior Editor

PLOS Biology

REVIEWS:

Reviewer #1: The authors examined the human brain activity evoked while playing games and found several distinctive functional domains in cortical and subcortical areas. The approach is unique and powerful and can assess interactive and self-relevant cognitive components in a controlled manner.

My concern is about their claim regarding synchronization. The authors argued about the existence of recurrent and dynamic synchronization across functional components (Fig. 1). However, given the slowness of fMRI signals, the results cannot provide strong evidence regarding the underlying interactive process. Instead, a more straightforward interpretation is that each brain region is involved in multiple distributed cognitive processes, as reported in recent literature (e.g., Shine et al., 2019; Nakai and Nishimoto, 2020) without assuming recurrent synchronization. The authors should elaborate on potential alternative interpretations and clarify what the current results can and cannot resolve.

Optionally, it would be great if the authors could provide a discussion on whether there is a sort of emotion present in a game-playing AI agent (e.g., Mnih et al., 2015; Vinyals et al., 2019), which has internal states that correspond to the variables discussed in this manuscript.

Shine JM, Breakspear M, Bell PT, et al. Human cognition involves the dynamic integration of neural activity and neuromodulatory systems. Nat Neurosci. 2019;22(2):289-296.

Nakai T, Nishimoto S. Quantitative models reveal the organization of diverse cognitive functions in the brain. Nat Commun. 2020;11(1):1142.

Mnih V, Kavukcuoglu K, Silver D, et al. Human-level control through deep reinforcement learning. Nature. 2015;518(7540):529-533.

Vinyals O, Babuschkin I, Czarnecki WM, et al. Grandmaster level in StarCraft II using multi-agent reinforcement learning. Nature. 2019;575(7782):350-354.

Reviewer #2: In this paper, Leitão and colleagues describe research that explores how appraisal, motivational, expressive, and physiological components contribute to the emergence of emotion. The research uses an interesting combination of a novel experimental paradigm, fMRI, and computational approaches to show the neural correlates of these components during emotional experiences, and how they interact. This manuscript has multiple strengths: it focuses on a timely and under-studied problem, it is clearly theoretically motivated, it includes multiple measures of emotion in addition to fMRI, and the statistical procedures appear appropriate. However, the significance of the results are somewhat difficult to appreciate as interpretation of findings relies on reverse inference and there are a few critical methodological and conceptual issues, as detailed below.

Major issues:

1. Does the experimental manipulation have external validity? It seems like the types of negative emotions the CPM tries to explain (e.g., responding to a dangerous, life-threatening event) are quite different than playing a video game in the fMRI environment. To what degree are the self-report and brain measures valid indicators of "real life" emotions? 

2. Interpretation of different brain systems largely relies on reverse inference, where the functional relevance of different brain regions is assumed rather than tested. Brain regions associated with increased coping potential were interpreted to reflect enhanced motor action control and planning during power. What are the relationships between brain activity associated with the pattern of brain activity observed with coping potential? Are they associated with measured differences in behavior in this way? The authors also note that brain regions are modulated by Goal Conduciveness are reminiscent of networks often reported for positive/rewarding and negative/aversive stimuli. How well does brain activity predict variation in self-reported valence (e.g., for trials similar to those used to assay subjective feelings post-scanning)? Linking component processes to better-established behaviors could better corroborate the findings.

3. Although some degree of neural synchronization takes place, it is not clear what drives this synchrony. Could it be explained by classic views of emotion where an appraisal of a particular event in the game triggers changes in autonomic physiology, expression, motivation, and feelings? Does the CPM make predictions about brain synchrony that differ from other theoretical views? How is recurrence needed to explain brain activity during the task? Do componential views offer insight here? In the absence of more direct comparisons between accounts, it is difficult to tell what we have learned from this interesting study.

Minor points.

1. The beginning of the introduction frames the paper around identifying essential neural circuits for emotion, but this is presumably not being tested with fMRI.

2. The use of liquid states machines is interesting, but could perhaps be described more extensively for readers unfamiliar with the approach

3. The discussion (and some of the introduction) suggests that most work on emotion does not focus on actual behavior. It is surprising that past work in this space is not cited or discussed, especially related work by Dean Mobbs using a similar manipulation with fMRI.

---

## [Decision Letter · Decision Letter 2]

16 Jul 2020

Dear Dr Leitao,

Thank you for submitting your revised Research Article entitled "Brain networks of emotions in sync: computational imaging of video-game playing" for publication in PLOS Biology. I have now obtained advice from the original reviewers and have discussed their comments with the Academic Editor. 

In light of the reviews (below), we are pleased to offer you the opportunity to address the remaining points from the reviewer 1 in a revised version that we anticipate should not take you very long. We will then assess your revised manuscript and your response to the reviewers' comments and we may consult the reviewers again.

We expect to receive your revised manuscript within 1 month.

**IMPORTANT - SUBMITTING YOUR REVISION**

*Resubmission Checklist*

*Published Peer Review*

*PLOS Data Policy*

*Blot and Gel Data Policy*

Sincerely,

Gabriel Gasque, Ph.D., 

Senior Editor

PLOS Biology

REVIEWS:

Reviewer #1: The authors provided some more details and discussions that support their original arguments regarding recurrent or feedback components. In particular, the additional Table S10 and the related discussion seem to be critical.

That being said, I am still not fully convinced, partly because they did not provide interpretable details of their methodology. The following information and control analysis will provide more concrete support:

(I) They should provide more methodological details on (A) feedback vs. (B) non-feedback models in Table S10. They could modify Figures 6/S2 to illustrate A and B. It is unclear what "lambda" indicates, as they do not show it in any equation, diagram, etc. It is also unclear what "rough cross-validation" means.

(II) Please provide the following negative and positive control simulation analyses:

(1) Prepare mock fMRI/behavior model signals with or without feedback components.

(2) Add correlated noise to (1).

(3) Apply temporal low-pass filtering to (2).

(4) Run the analysis pipeline (Figure S2?) to (3).

(5) Test analyses in Table S10.

If the simulation analyses above can detect the presence and absence of feedback components, that should be convincing evidence that their pipeline works to examine the difference, even using challenging fMRI data.

Reviewer #2: The authors have done an excellent job of addressing all points raised during review. This work is exceptional for its combination of theory, unique experimental design, and computational approaches. I hope it inspires similar research in this area.

---

## [Editor Report · Decision Letter 3]

20 Aug 2020

Dear Dr Leitao,

Thank you for submitting your revised Research Article entitled "Brain networks of emotions in sync: computational imaging of video-game playing" for publication in PLOS Biology. I have now discussed your revision with the Academic Editor, and I'm delighted to let you know that we're now editorially satisfied with your manuscript. 

However, we would like you to consider changing the name of the article to convey the central biological message of your findings. We suggest the following title, but would be happy to discuss alternatives:

"Computational imaging of video-game playing shows dynamic synchronization of cortical and sub-cortical networks of emotions"

In addition, before we can formally accept your paper and consider it "in press", we also need to ensure that your article conforms to our guidelines. A member of our team will be in touch shortly with a set of requests. As we can't proceed until these requirements are met, your swift response will help prevent delays to publication. Please also make sure to address the data and other policy-related requests noted at the end of this email.

*Copyediting*

*Published Peer Review History*

*Early Version*

*Submitting Your Revision*

Sincerely,

Gabriel Gasque, Ph.D.,

Senior Editor,

ggasque@plos.org,

PLOS Biology

ETHICS STATEMENT:

-- Please indicate within your manuscript if your protocol approved by the Research Ethics Committee of the Geneva University Hospital adhered to the Declaration of Helsinki or any other specific national or international ethical guidelines.

-- Please include the ID number of your approved protocol.

DATA POLICY:

Note that we do not require all raw data. Rather, we ask for all individual quantitative observations that underlie the data summarized in the figures and results of your paper. For an example see here: http://www.plosbiology.org/article/info%3Adoi%2F10.1371%2Fjournal.pbio.1001908#s5

These data can be made available in one of the following forms:

Regardless of the method selected, please ensure that you provide the individual numerical values that underlie the summary data displayed in the following figure panels: Figures 3, 4, 5, 7, S1, S3, S4, S5, S6, and S7. 

Please also ensure that each figure legend in your manuscript include information on where the underlying data can be found and ensure your supplemental data file/s has a legend.

---

## [Editor Report · Decision Letter 4]

6 Oct 2020

Dear Dr Leitão,

On behalf of my colleagues and the Academic Editor, Kevin S LaBar, I am pleased to inform you that we will be delighted to publish your Research Article in PLOS Biology. 

Early Version

PRESS 

Kind regards,

Gabriel Gasque,

Senior Editor

PLOS Biology